# Extended Exposure Topotecan Significantly Improves Long-Term Drug Sensitivity by Decreasing Malignant Cell Heterogeneity and by Preventing Epithelial–Mesenchymal Transition

**DOI:** 10.3390/ijms24108490

**Published:** 2023-05-09

**Authors:** Joshua T. Davis, Taraswi Mitra Ghosh, Suman Mazumder, Amit Mitra, Richard Curtis Bird, Robert D. Arnold

**Affiliations:** 1Department of Drug Discovery and Development, Auburn University, Auburn, AL 36849, USA; tmitragh@auburn.edu (T.M.G.); szm0006@auburn.edu (S.M.); akm0060@auburn.edu (A.M.); 2Department of Urology Research, Brigham and Women’s Hospital, Harvard Medical School, Boston, MA 02215, USA; 3UAB O’Neal Comprehensive Cancer Center, University of Alabama at Birmingham School of Medicine, Birmingham, AL 35233, USA; 4Center for Pharmacogenomics and Single-Cell Omics (AUPharmGx), Harrison College of Pharmacy, Auburn University, Auburn, AL 36849, USA; 5Department of Pathobiology, College of Veterinary Medicine, Auburn University, Auburn, AL 36849, USA; birdric@auburn.edu

**Keywords:** oncology, alternative dosing, resistance, heterogeneity, transcriptomics, spheroid model, long-term exposure

## Abstract

Maximum tolerable dosing (MTD) of chemotherapeutics has long been the gold standard for aggressive malignancies. Recently, alternative dosing strategies have gained traction for their improved toxicity profiles and unique mechanisms of action, such as inhibition of angiogenesis and stimulation of immunity. In this article, we investigated whether extended exposure (EE) topotecan could improve long-term drug sensitivity by preventing drug resistance. To achieve significantly longer exposure times, we used a spheroidal model system of castration-resistant prostate cancer. We also used state-of-the-art transcriptomic analysis to further elucidate any underlying phenotypic changes that occurred in the malignant population following each treatment. We determined that EE topotecan had a much higher barrier to resistance relative to MTD topotecan and was able to maintain consistent efficacy throughout the study period (EE IC50 of 54.4 nM (Week 6) vs. MTD IC50 of 2200 nM (Week 6) vs. 83.8 nM IC50 for control (Week 6) vs. 37.8 nM IC50 for control (Week 0)). As a possible explanation for these results, we determined that MTD topotecan stimulated epithelial–mesenchymal transition (EMT), upregulated efflux pumps, and produced altered topoisomerases relative to EE topotecan. Overall, EE topotecan resulted in a more sustained treatment response and maintained a less aggressive malignant phenotype relative to MTD topotecan.

## 1. Introduction

Metronomic or extended exposure (EE) dosing of chemotherapeutics was first introduced as an antiangiogenic therapy by Dr. Folkman, Dr. Browder, and Dr. Kerbel in 2000 [1,2,3]. In contrast to maximum tolerable dosing (MTD), which is usually administered as a large single-dose or a short course of therapy at a level just below life-threatening toxicities, EE dosing is usually administered more frequently at much lower doses and at a cumulative dose that may be at or significantly below MTD. It was hypothesized that EE dosing would more effectively target endothelial cells and would prevent the reflexive regeneration of endothelial cells that can occur during the drug-free periods of conventional therapy. It was also thought that endothelial cells would not develop resistance because they were genetically stable. The antiangiogenic mechanism of EE chemotherapy would later be confirmed with multiple agents and in multiple cancer types; however, malignant cells proved more versatile than anticipated and, in many instances, developed drug resistance [3,4,5,6,7]. It was later revealed that the mechanism of action of EE therapy was likely multimodal. Some major mechanisms that have been identified include inhibiting angiogenesis, normalizing existing vasculature, activating the immune system, and inducing tumor dormancy and senescence, but complete understanding is lacking [3,5,8,9].

We first investigated EE dosing of topotecan using an in vivo xenograft model of subcutaneously implanted human metastatic prostate adenocarcinoma (PC3) cells. EE topotecan, which was administered using a subcutaneously implanted osmotic pump, was compared to MTD topotecan, which was administered as a bolus dose using tail vein injections. In this experiment, EE topotecan significantly reduced tumor growth relative to MTD topotecan. Importantly, we used an athymic mouse model, which should have limited most immune-related mechanisms of EE topotecan. We also did not find any significant differences in the tumor vasculature density for any treatment group [10,11,12]. In our in vitro experiments, when maintaining equivalent cumulative exposure, clinically meaningful changes to the IC50 could not be produced over a 72 h timepoint (MTD IC50 190 nM, EE IC50 177 nM). Therefore, seemingly, the three major mechanisms of EE dosing (angiogenesis, immunity, and direct effects) could not adequately explain our results.

Thus, EE topotecan, somewhat paradoxically, produced similar short-term efficacy and greater long-term efficacy relative to MTD topotecan. We determined the most plausible explanation for these seemingly incongruous results was a change in drug sensitivity over time by the underlying malignant cell population, or, said in another way, that MTD topotecan led to rapid regimen crippling resistance, which was attenuated by EE topotecan. This article describes the methods used and the evidence obtained to determine if alternative dosing schedules of chemotherapeutics can change the phenotypic characteristics of surviving cell populations, thus altering their long-term sensitivity.

## 2. Overview of Epithelial–Mesenchymal Transition (EMT)

EMT is the phenotypic transition of a cell from an epithelial-like state to a mesenchymal-like state. Typical characteristics of epithelial cells include apical–basal polarity, structural cell–cell connections with adherens junctions, tight junctions, and desmosomes, and connection to the basement membrane through hemidesmosomes. Mesenchymal cells, on the other hand, typically lose cell–cell connections, possess anterior–posterior polarity, and have strong migratory properties. EMT is induced during three main physiological events: embryonic development, tissue regeneration, and cancer progression [13,14]. Cells undergoing EMT lose classic epithelial markers such as E-cadherin or EpCAM, decrease the production of mucins and other epithelial matrix molecules, and shed adhesion molecules. These cells also begin to increase the production of mesenchymal markers such as N-cadherin, vimentin, and fibronectin. Importantly, this transition is gradual, and cells usually fall within a range between highly epithelial to highly mesenchymal. This process is also reversible. Mesenchymal-like cells can become more epithelial-like through mesenchymal–epithelial transition (MET). EMT can be triggered by many different environmental factors such as hypoxia, cytokines, growth factors, and therapeutic agents. The most common regulatory factors for EMT include SNAIL (SNAI1), SLUG (SNAI2), TWIST1, TWIST2, ZEB1, and ZEB2. Cancer cells that have undergone EMT display other characteristics such as increased stemness, increased migratory potential, increased chemoresistance, and decreased immune sensitivity [13,14,15]. Overall, EMT shifts cancer cells into a more aggressive, durable, and resistant phenotype. Figure 1 summarizes the primary features and processes of EMT.

## 3. Results

### 3.1. Comparing the Long-Term Potency of EE and MTD Topotecan

First, we determined whether different dosing schedules of topotecan could alter long-term topotecan potency. To do this, we used a 3D spheroidal model of PC3 cells that can be maintained and treated for weeks to months to assess weekly changes in IC50 potency. Each dose was given as a weeklong exposure with intervening drug-free intervals (in 2D). The intervening drug-free intervals allowed us to accurately replate cells for each IC50 assay. This prevented an ever-increasing week-to-week sample variability that could occur in a strict longitudinal assay. The EE dose (14.3 nM) was given daily at 1/7^th^ the MTD dose (100 nM), which was given as a bolus on day 0. The total cumulative exposure of topotecan for each treatment was equivalent throughout the experiment. We also included untreated spheroids, which served as a control. Each treatment was administered for 6 weeks, which amounted to roughly 3–4 months of total study duration when accounting for the drug-free intervals. After the first treatment, each treatment group was maintained as a separate cell line for the remainder of the experiment. After each week of exposure, an IC50 assay was performed, and samples were stored for future genomic, transcriptomic, and proteomic analysis. A plot of the long-term topotecan potency can be found in Figure 2 and sample images of the spheroids after 5 weeks of exposure can be found in Figure A1. For the complete study period, relative to the initial untreated control sample, the IC50 of topotecan increased 2.21-fold for the control spheroids, increased 1.44-fold for the EE spheroids, and increased 58.3-fold for the MTD spheroids.

### 3.2. Determining the Impact of Different Dosing Strategies on Population Heterogeneity

Intratumor heterogeneity is a major cause of drug resistance and can result in worse clinical outcomes for patients. Heterogenous populations are more genetically and phenotypically diverse, which increases the probability that a resistance-inducing phenotype or mutation is present in the underlying cell population. Heterogenous populations also possess a more variable exposure response profile at an individual level, which may protect some cells from death and allow further resistance to develop over time [21,22,23]. Because of these factors, it was important to determine the impact of MTD and EE treatments on the underlying heterogeneity of the population as this could affect drug potency over time. We analyzed our scRNAseq data using t-SNE, which is a nonlinear dimensionality reduction technique that arranges similar objects as nearby points and dissimilar objects as distant points [24]. In this analysis, we used samples that were obtained from the IC50 study at different timepoints. In particular, we compared 2D samples taken on the last day of their drug-free interval, just prior to reseeding for the 6th week of treatment to 3D samples taken after the final day of the 6th week of treatment (see Figure A2). This helped us understand the impact of the 3D model on heterogeneity (a/b), to understand the immediate impact of drug treatment on heterogeneity (c/d and e/f vs. a/b), and to understand the long-term impact of drug treatment on heterogeneity (c/e vs. a) (Figure 3). The control 2D (a) and 3D (b) graphs show a modest increase in heterogeneity in the 3D sample. The MTD-treated 3D sample (d) is more heterogeneous than the 3D control sample (b), is more heterogeneous than the 2D drug-free interval MTD sample (c), and is more heterogeneous than the EE-treated 3D sample (f). The EE-treated 3D sample (f) is more uniform than the 2D drug-free interval EE sample (e) and is relatively similar to the 3D control sample (b). The MTD (c) and EE (e) drug-free interval 2D samples are more heterogeneous than the 2D drug-free interval control sample (a).

### 3.3. Evaluating the Underlying Molecular Causes of MTD-Induced Drug Resistance

We used RNAseq to identify underlying transcriptomic differences between the EE- and MTD-treated cells. Out of an initial list of 1000 genes selected based on the lowest *p*-value, 189 genes were selected for further analysis. A total of 51 out of the 189 genes did not have a well-defined function or had limited information available in the literature (see Figure 10). After additional screening criteria, 94 total genes were selected for analysis. The overall expression pattern for the complete list of 94 genes is shown in the heatmap (Figure 4). These genes have well-known functions in cell adhesion, tumor suppression, or malignancy progression, or are well-known epithelial markers. The heatmap (Figure 4) revealed a significant deviation from control for the MTD-treated cells over time. During the first week (days 1, 3, 7), most genes were not differentially expressed relative to the control cells (grey boxes), indicating transcriptomic similarity to the control cells. However, *CXCL8* is a notable exception for the MTD-treated cells, which generated a 14.5-fold change relative to the control cells on day 1. The EE-treated cells generated a 1.5-fold change for *CXCL8* on day 1. After 2 weeks of treatment, most genes from the MTD- and EE-treated cells were significantly (*p* < 0.05) different to the control cells but were not drastically different. At this point, the MTD- and EE-treated cells looked relatively similar. Over the next few weeks, however, the MTD-treated cells began to significantly diverge from both the control group and the EE-treated cells. In many instances, genes that were perturbed in week 3 became directionally more perturbed throughout the experiment with some genes registering a change of over 300-fold by week 6.

Figure 5, Figure 6 and Figure 7 highlight a few specific genes associated with EMT and provide a graph of the fold change over time for the MTD- and EE-treated cells. Only points with a statistically significant (*p* < 0.05) change relative to the control group were included in each graph. In Figure 5, we identify important epithelial markers that are significantly downregulated during EMT. Important gene types in this list are claudins (*CLDN7*), adhesion molecules (*CDH1*/E-Cadherin, *EPCAM*, or *LAMB3*), mucins (*MUC2*, *MUC5AC*, or *MUC6*), keratins (*KRT7*, *KRT80*), and *PATJ*, which regulates both tight junctions and cell polarity [25]. The orange line for all graphs represents fold change over time relative to the control for the MTD-treated cells. Almost every gene identified in this graph was significantly downregulated in the MTD-treated cells. In contrast, the expression profile of the EE-treated cells (blue line) was relatively stable throughout the experiment. Figure 6 highlights genes that are known to regulate EMT. We included genes that were not included in the heatmap (Figure 4) to help provide a comprehensive view of the EMT transition for these cells (asterisks). *ESRP1*, *ESRP2*, *GRHL2*, *NOTCH3*, *OVOL1*, *ZEB1*, and *CXCL8* were altered most significantly by the MTD treatment. Similar to the epithelial markers listed in Figure 5, gene expressions from the EE-treated cells were mostly stable throughout the experiment. The remaining genes (*QKI*, *RBFOX2*, *SRSF1*, *TCF3*, and *YAP1*) did not have many significant data points for either treatment, and the few data points that were significant to the control were not significantly different between each treatment group. In Figure 7, we show the long-term gene expression profile of the MTD- and EE-treated cells for many well-known mesenchymal markers. It should be noted that only *CDH11* met the inclusion criteria for our gene list in Figure 4. *CDH2* (N-cadherin), *CTNNB1* (beta-catenin), and *S100A4* had many missing (insignificant) data points, but the data points available were not significantly different from each other or from the control. *FN1* (fibronectin), *ITGA5* (integrin-α5), *LAMA5* (laminin 5), and *VIM* (vimentin) were not significantly altered by either treatment group relatively, and were not significantly different from the control cells.

### 3.4. Assessing Whether Alternative Administration Schedules Can Alter Efflux Pump and Topoisomerase Expression

The effect of alternative dosing strategies on efflux pump and topoisomerase expression patterns was determined because of their potential to cause topotecan drug resistance. ScRNAseq provided a more in depth understanding of the individual expression patterns within the cancer population. In this experiment, we used the same sample set that was used in Figure 3. These samples are identified with letters in Figure 8 and Figure 9. The efflux pump expression data can be found in Figure 8. When comparing the 2D control group (a) to the 3D control group (b), an increase in the density of efflux pump expression can be seen with many more cells from the 3D control group expressing efflux pumps. The 3D control group increased the expression of the *ABCA7*, *ABCC3*, and *ABCC4* efflux pumps. Interestingly, the 2D MTD (c) and EE I drug-free interval populations also showed increased efflux expression frequency with both groups highly expressing *ABCC3*, *ABCC4*, *ABCC5*, and *ABCG1*. The treated 3D MTD (d) sample also increased the density of cells expressing efflux pumps with *ABCC3* and *ABCC5* being the most prominent. On the other hand, the 3D EE-treated sample (f) did not display increased efflux pump expression relative to the 2D control sample. In fact, it could be argued that the EE-treated sample reduced efflux pump expression relative to the control 2D sample.

Next, because topoisomerases are the main target of topotecan (specifically topoisomerase I), we wanted to assess whether alternative dosing strategies could alter the topoisomerase expression patterns of our treatment populations, which could play a role in drug resistance (Figure 9). Although there are several insights that can be identified in these data, one of the most striking sample characteristics is the relative expression of topoisomerase I (*TOP1*) to topoisomerase II (*TOP2*) and topoisomerase III (*TOP3*). The control 2D (a) sample produced more *TOP1* and less *TOP2* with limited *TOP3* expression. The control 3D (b) sample showed increased *TOP2* relative to *TOP1*, and limited *TOP3*. The 2D drug-free interval MTD sample (c) increased its *TOP2* expression relative to *TOP1* but also increased its *TOP3* expression. The 2D drug-free interval EE sample (e) also showed a similar increase in *TOP2* and *TOP3* expression relative to *TOP1*. The 3D MTD-treated sample (d) increased its *TOP2* and *TOP3* expression relative to *TOP1* but had the lowest amount of *TOP1* of all samples. The 3D EE-treated sample (f) displayed much higher *TOP1* expression relative to the MTD-treated sample and rivaled even the control 2D group. This sample also produced some *TOP2* but limited *TOP3*.

## 4. Discussion

Metronomic or extended exposure dosing of oncologic agents is a relatively new paradigm with the potential to improve efficacy and reduce toxicity in some patients. To date, this treatment modality has demonstrated the ability to impact angiogenic and immunologic targets. We investigated a potentially novel mechanism impacting drug resistance. In Figure 2, we used a long-term spheroidal model of PC3 cells to investigate the potency of topotecan over time after multiple weeks of treatment with either MTD- or EE-dosed topotecan. After 6 full weeks of drug exposure or approximately 3–4 months total, we demonstrated demonstrably decreased potency by the MTD-treated cells. On the other hand, the EE-treated cells maintained potency in line with the control cells. These data suggest that drug dosing can have a substantial impact on the underlying cell populations, which can significantly affect efficacy. This also underscores the need to better understand how therapeutics impact tumor cells and whether, long-term, we are creating more aggressive and resistant tumor cells to acutely reduce tumor volume. It also suggests that drug screening and selection should occur in longer-term model systems to appropriately identify treatments that can achieve sustained success. If we were to convert the potency data for MTD topotecan into a clinical scenario, it would suggest that after a single treatment, a patient would require approximately 4–5× the initial dose to have a similar impact on tumor cells. After 5 weeks of treatment, a patient would require 40× the initial dose, which is clinically unfeasible. If a treatment cannot eliminate tumor cells completely, which is currently true for almost all oncologic therapeutics, then maintaining a sensitive cancer cell population is vitally important.

To appropriately evaluate the impact of drug dosing on treatment resistance, we required a model system with adequate exposure duration, variable individual cell exposure through physical barriers and treatment gradients, and increased intratumor-like heterogeneity through added model complexity. Using scRNAseq (Figure 3), we evaluated the impact of each treatment as well as the model system on population heterogeneity. Although the 3D spheroid did show increased heterogeneity relative to the 2D cells, it was less than initially expected. This was most likely caused by carryover effects from prior exposure to the spheroid model, similar to what was shown in the EE and MTD drug-free interval samples. Regardless, the most striking results from this experiment were found in the EE- and MTD-treated 3D samples. The MTD-treated 3D samples significantly increased heterogeneity, even relative to the elevated level of heterogeneity found in the underlying 2D drug-free interval MTD cells. Although the 2D drug-free interval EE cells were found to possess similar heterogeneity to the 2D drug-free interval MTD cells, the EE-treated 3D cells displayed significantly reduced heterogeneity. These cells seemed to phenotypically align in response to a more drawn-out topotecan exposure. This result highlights the potential role of EE topotecan as a modulator of cancer cell heterogeneity. Because increased heterogeneity has been shown to increase drug resistance and lead to poor clinical outcomes, reducing the genetic diversity of cancer cells prior to therapy might increase the efficacy of combination therapeutics [21,22,23]. These results also highlight the need to further understand the impact of other therapeutics on cancer cell heterogeneity. Doing so may permit the ranking of therapeutics based on their impact on cellular heterogeneity, allowing clinicians to select more effective regimens.

To further understand why MTD-dosed topotecan led to such a divergent potency response, we used RNAseq to help identify the top differentially expressed genes from the EE and MTD-treated cells. A summary of these results is presented as a heatmap (Figure 4). For this set of genes, both treatment groups remained relatively stable after the first and second weeks of exposure, however, by week 3, significant changes to the MTD-treated cells could be seen, which further progressed over weeks 4 and 5. These changes also correlated well with our IC50 data, which supported further probing to determine each gene function and to determine if a mechanism of resistance could be identified.

A pattern emerged and EMT appeared most likely to cause the potency differences found between EE and MTD topotecan-treated cells. Genes that support this hypothesis have been identified in Figure 5, Figure 6 and Figure 7. EMT usually involves the loss of epithelial markers and the gain of mesenchymal markers. The mRNA isolated from the MTD-treated cells showed significant downregulation of keratins, which are found in cornified and stratified epithelial cells and are known to be inhibited in EMT [17,18,26]. They also downregulated each of the secreted mucins (*MUC2*, *MUC5AC*, *MUC5B*, and *MUC6*). Of these, MUC2 has been found to be most important in oncologic disease. Loss of MUC2 expression in mice is associated with increased proliferation and survival of intestinal epithelial cells and is associated with invasive adenocarcinomas [27]. The MTD-treated cells also downregulated a substantial number of genes associated with cell adhesion, which is a common sign of EMT. CDH1 or E-Cadherin is the most common EMT-associated adhesion protein and is almost always downregulated during EMT [13,14,15,17,18]. The MTD-treated cells reduced *CDH1* expression 68.5-fold relative to the control cells compared to a 3.6-fold decrease by the EE-treated cells. The MTD-treated cells also downregulated a number of other adhesion genes such as *EPCAM* [28,29], *CEACAM5* [30], *GJB3* [31], *TJP3* [32], *LAD1* [33], *MPZL2* [34], and *LSR* [35].

Furthermore, other genes associated with epithelial cells were significantly perturbed by the MTD-treated cells. For instance, EPHA1 is associated with ephrin signaling, which helps regulate the actin cytoskeleton, and is localized to epithelial junctions by E-cadherin. The loss of E-cadherin by the MTD-treated cells caused the downregulation of the *EPHA1* gene, further supporting the loss of an epithelial phenotype [36]. The *ERBB3* gene, which encodes a well-known growth factor receptor in cancer also showed significant downregulation in MTD treated cells [32]. The *MYO5B* gene, which encodes a protein associated with apical–basolateral polarization, is also downregulated [37]. Similarly, *PATJ*, a gene that encodes a protein that regulates tight junction formation and polarization, is downregulated [25]. The *PRSS8* gene produces a glycosylphosphatidylinositol-anchored epithelial extracellular membrane serine protease prostasin, which is expressed abundantly in normal epithelial cells and is essential for terminal epithelial differentiation, but is downregulated by MTD-treated cells. Downregulation of this protein has been associated with EMT in human bladder carcinomas [38] and is associated with increased growth and metastasis in hepatocellular carcinoma [39]. The *LCN2* gene produces a protein within the lipocalin superfamily and has been found to be expressed highly in early-stage colorectal cancer, but is downregulated significantly in metastatic or advanced-stage colorectal cancer, which may suggest that the MTD-treated cells are not only transitioning into a more mesenchymal phenotype but are also significantly more aggressive [40]. The loss of any single epithelial gene would not support an EMT hypothesis; however, the consistent downregulation of many epithelial genes simultaneously indicates that the MTD-treated cells are most likely undergoing EMT, while the EE-treated cells appear to remain relatively phenotypically stable.

We also analyzed established EMT regulatory genes (Figure 6). Of these, Snail (*SNAI1*), Slug (*SNAI2*), *TWIST1*, *TWIST2*, *ZEB1*, and *ZEB2* are the most well-known; however, many of these factors do not appear to play a significant role in the EMT transition of the MTD-treated cells based on our RNAseq expression data [13,14,15,17,18]. Although CXCL8 is does not strictly play a regulatory role in EMT and is more accurately classified as an EMT trigger, we included its expression profile in this figure to highlight its initial burst of expression during the first day of exposure and consistent decline in expression during the remainder of the experiment. In addition to *CXCL1* to a lesser extent (Figure 4), *CXCL8* is one of the few genes (3 in total using our criteria) with significant early altered expression relative to the control. CXCL8 is known to promote proliferation, inhibit apoptosis, increase heterogeneity, and stimulate EMT [41]. Additionally, elevated CXCL8 expression is correlated with high Gleason scores and elevated PSA [42]. Based on our data and CXCL8’s known role in EMT, it is likely that elevated CXCL8 expression is an important early trigger of MTD topotecan-induced EMT and drug resistance. *ZEB1* was not originally identified based on the selection criteria; however, after further evaluation, it appears to be significantly altered by the MTD-treated cells and is not consistently altered by the EE-treated cells. Further supporting ZEB1’s role, *TRIM29* was downregulated in MTD-treated cells and its protein is associated with increased ZEB1 expression and EMT in cervical cancer cells [43]. Additionally, MTD-treated cells downregulated *FXYD3*, which was found to be downregulated in mammary epithelial cells because of TGFβ and ZEB1 signaling, supporting ZEB1’s role in the EMT of MTD-treated cells [44]. NOTCH3 is another important regulator associated with chemotherapy resistance in esophageal cancer cells when downregulated. In this study of esophageal cancer, silencing NOTCH3 resulted in increased production of VIM and resulted in increased chemotherapy resistance [45]. In another study, NOTCH3 was found to inhibit EMT in breast cancer by activating downstream transcription complexes [46]. Our results also highlight the important role of *NOTCH3* in regulating EMT as it was one of the first regulatory genes to become significantly downregulated (41.2-fold by week 3) by the MTD-treated cells. ESRP1 and ESRP2 are epithelial splicing regulatory proteins that regulate alternative splicing events associated with epithelial phenotypes and are significantly downregulated during EMT [47]. Further supporting this finding, *OVOL1* was significantly downregulated in MTD-treated cells. OVOL1 has been found to induce mesenchymal–epithelial transition (MET) by upregulating ESRP1. OVOL1 is also a part of a regulatory feedback loop with ZEB1. Thus, its downregulation correlates with a downregulation of ESRP1 and an upregulation of ZEB1 [48]. Lastly, a recent article highlights the EMT suppressor role of Grainyhead-like 2 in ovarian cancer cells. *GRHL2* was significantly downregulated by the MTD-treated cells (279-fold by week 5), and many of the genes identified in the article were also altered by the MTD-treated cells (*KRTs*, *GRHL2*, *ESRP1/2*, *EPCAM*, *CDH1*, *CDH3*, *ERBB3*, *ZEB1*, *CLDNs*, *PROM2*, *S100A14*, *SPRINT1*, *LAD1*, and *ST14*) [33]. GRHL2 knockdown was found to result in genome-wide epigenetic remodeling through increased methylation of CpG sites and through nucleosome remodeling. It was found that GRHL2 most likely regulated the CpG methylation of epithelial genes at its binding sites. It was also found that the GRHL2 knockdown would most likely cause an intermediate form of EMT [33]. Our results are consistent with their findings of the widespread knockdown of epithelial genes in response to a significant knockdown of GRHL2. The MTD-treated cells are likely in an intermediate stage of EMT as significant losses of epithelial markers are evident, but significant gains in mesenchymal markers are not evident (Figure 7).

The EE-treated cells did not lose *OVOL1*, *ESRP1*, *GRHL2*, or *NOTCH3* expression, nor did they significantly upregulate *ZEB1* expression consistently. These cells also did not significantly upregulate mesenchymal markers and did not significantly downregulate their epithelial markers. Importantly, long-term fractionated dosing of topotecan appeared to prevent EMT within these cells while still maintaining efficacy, which prevented EMT-induced drug resistance. This conclusion is further supported in Figure 8 and Figure 9, which show that EE-treated cells maintained similar expression patterns of efflux pumps and topoisomerase genes compared to control cells. On the other hand, a greater proportion of MTD-treated cells expressed efflux pumps and alternative topoisomerase genes. Each of these mechanisms could reduce the exposure or efficacy of topotecan and likely contribute to the reduction in IC50 potency (Figure 2).

We demonstrated that alternative dosing strategies can have a substantial impact on underlying cell populations, which can directly affect treatment outcomes. These results also support the need for frequent genetic testing when administering oncologic therapeutics to quickly identify failed therapies and avoid harming patients. Finally, these results call into question the use of short-term efficacy models as drug-screening tools and support the need to better understand the temporal impact of oncologic medications on surviving cell populations.

## 5. Materials and Methods

### 5.1. Cell Line and Cell Culture

The human prostate cancer (PC3) cell line was obtained from ATCC and was maintained as monolayers in complete medium using F12K (Corning, Corning, NY, USA) and 10% (*v*/*v*) FBS (Hyclone, Logan, UT, USA) at 37 °C in a 5% CO_2_ atmosphere using a Heracell bios 160i incubator (Thermo Fisher Scientific, Waltham, MA, USA). During the experiment, the PC3 cell line was divided into multiple sub-cell lines according to the treatment group, which will be described in greater detail below. Each of these sub-cell lines were treated as a unique cell line (separate flasks, no mixing, etc.) throughout the experiment using the same methods described above.

### 5.2. Spheroid Formation

The spheroid protocol was largely adapted from a high-throughput liquid overlay technique developed by Metzger et al. [49]. This technique rapidly generates many spheroids with minimal incubation time (24 h), which is necessary for drug-screening protocols. Briefly, 96-well U bottom plates (Grenier bio-one, Monroe, LA, USA) were coated with a 1.2% (*w*/*v*) poly-HEMA (Sigma Aldrich, St. Louis, MO, USA) solution in 95% (*v*/*v*) ethanol. This solution was produced by incubating poly-HEMA crystals overnight with a magnetic stir rod at 80 °C to ensure full dissolution. The poly-HEMA solution was kept warm throughout the coating process to prevent precipitation during the evaporation step. A 60 μL volume of the poly-HEMA solution was added to each well and the plates were heated using a hot plate (VWR, Radnor, PA, USA). Plates were left on the hot plate for approximately 1 h with the lid raised to evaporate the ethanol. Plates were then sealed using Parafilm (Bemis, Neenah, WI, USA) for future use. After cells were passaged and placed into a separate conical tube, they were mixed thoroughly, and a small sample was removed for counting using a TC10 automated cell counter (Biorad, Hercules, CA, USA). A minimum of two counts were taken per cell line to ensure accurate counts for cell seeding. Cells were diluted to achieve a concentration of 50,000 cells per mL and placed on ice. A total of 2.5% (*v*/*v*) of Matrigel (Corning, Corning, NY, USA) was added to the cell suspension using an ice-cold syringe and needle. The cells were then plated using 100 μL of the cell suspension to attain 5000 cells per well. The plates were then centrifuged at 400 g for 5 to 10 min at 4°C. This protocol rapidly generates fully formed spheroids within 24 h for the PC3 cell line.

### 5.3. Dosing and Spheroid Handling

Two days after initial seeding and spheroid formation, an additional 100 μL of media +/− drug was added to achieve a total volume 200 μL for the remainder of the experiment. On days 3 and 5, a media exchange was performed by removing 100 μL of media per well and replacing it with 100 μL of fresh complete media +/− drug. Limiting the media exchanges and leaving some residual, old media prevented spheroid loss throughout the experiment. On off-media exchange days, 10 μL of media was removed and replaced with 10 μL of media or treatment solution according to the treatment group. Dosing of topotecan (Chempac, Synder, TX, USA) and docetaxel (Fluka, Charlotte, NC, USA) occurred using 20× concentrated solutions, which could be directly spiked into the wells at 10 μL in 190 μL of media. The conventional (MTD) treatment was given as a bolus dose on day 0. Metronomic or EE treatment was given daily as a fractionated dose at 1/7^th^ the MTD. The cumulative dose for the MTD and EE treatments was equal throughout the experiment. In total, there were 3 treatment groups: control, MTD topotecan, and EE topotecan. Topotecan dosing occurred at 100 nM during each week of therapy and occurred between 1 and 100,000 nM for the IC50 assays.

### 5.4. Study Protocol

Spheroids were generated and grown for approximately 2–3 days to allow size-dependent drug barriers to form. Spheroids were then dosed for a total of 7 days. During the first week of exposure, samples were taken for genomic and proteomic analysis on days 0, 1, 3, and 7. The remaining spheroids were saved for future weeks following digestion using Accumax (Innovative cell technologies, San Diego, CA, USA) for approximately 1 h until a single-cell suspension was achieved. A total of 3 treatment groups generated 3 unique cell lines that were maintained throughout the experiment: PC3-Control, PC3-EE-Topotecan, and PC3-MTD-Topotecan. The digested spheroids were grown in 2D for approximately 1–2 weeks until the cell population was replenished sufficiently to plate additional spheroids. Each cell population was then used to generate two groups of spheroids. One group (3D) was exposed to an additional week of treatment and one group (3D) was used to assess the resulting sensitivity of the drug (Topotecan) from the previous week(s) of drug exposure. After another full week of exposure, some spheroids were harvested for genomic and proteomic analysis, and some were digested to prepare for another week of exposure and analysis. This cycle was repeated throughout the experiment. A schematic is depicted (Figure A3 and Figure A4) to help illustrate the study protocol. For scRNAseq, we analyzed digested spheroids from week 5 (2D) and treated 3D samples from week 6. Week 5 samples had been grown in 2D for approximately 1 to 2 weeks in drug-free media before analysis.

### 5.5. Resazurin Assay (Cytotoxicity)

Resazurin was used to measure the mitochondrial activity of the cells as a surrogate for cell viability because the reductive conversion of resazurin to resorufin creates a water-soluble end product. This prevents the need for a solubilizing step, which would be untenable in a 3D format. Resazurin (Alfa Aesar, Haverhill, MA, USA) was made fresh for each assay at a 0.015% (*w*/*v*) concentration in PBS and was sterilized using a 0.22 μm filter. Before resazurin was added to the spheroids, the spheroids were moved from U bottom 96-well plates to flat bottom black, fluorescent plates (Grenier bio-one, Monroe, LA, USA). This was accomplished using a 1 mL pipette tip to move the spheroid and 100 μL media. Moving the spheroids increased the accuracy of the imaging and spectrophotometry. This also ensured that well volume variability from inconsistent evaporation dynamics that occur over the duration of the experiment would not alter the resorufin concentrations. Resazurin was added at a ratio of 10 μL per 100 μL of media and was incubated for 4 to 12 h with readings taken over time (2, 4, 6, 8, 12). Generally, 4-6 h was the most appropriate time point and achieved the lowest variation (CV values) with the greatest sensitivity and limited assay saturation. Fluorescent measurements for each plate were read using a Cytation 5 plate reader (BioTek, Winooski, VT, USA) with excitation set at 560 nm and emission set at 590 nm.

### 5.6. RNA Storage Protocol

Cells and spheroids were separated into individual microfuge tubes at approximately 1,000,000 cells/mL and washed 2× using PBS (Wards science, Rochester, NY, USA) and the Heraeus Fresco 21 microcentrifuge (Thermo Fisher Scientific, Waltham, MA, USA) set at 400 g and 4 °C for 10 min. Samples were maintained on ice for the duration of the protocol. PBS was aspirated and replaced with 300 μL of RNA later (Qiagen, Venlo, Netherlands). Samples were stored overnight (24 h) at 4 °C before moving to −80 °C for long-term storage. ScRNAseq samples were cryopreserved using 10% DMSO in complete media and stored in liquid nitrogen.

### 5.7. RNA Isolation

Total RNA was isolated from cultured cells and 3D spheroid model using standard RNA extraction kits (RNeasy Kits QIAGEN, Venlo, Netherlands). RNA concentration and integrity were estimated using a NanoDrop 2000 UV-Vis spectrophotometer (Thermo Scientific, Waltham, MA, USA), Qubit^®^ 2.0 Fluorometer (Invitrogen, Carlsbad, CA, USA), and Agilent 2100 Bioanalyzer (Applied Biosystems, Carlsbad, CA, USA). RNA integrity number threshold of eight was used for RNAseq analysis.

### 5.8. RNAseq

RNAseq libraries were constructed using Illumina TruSeq RNA Sample Preparation Kit v2. Libraries were then size selected to generate inserts of approximately 200 bp. RNA sequencing was performed on llumina’s NovaSeq next-generation high-throughput sequencing system using 150 bp paired-end protocol with a depth of more than 20 million reads per sample. The average quality scores were above Q30 for all libraries in both R1 and R2.

### 5.9. RNAseq Data Processing

RNAseq data were normalized, and fragments per kilobase million values were used in further analysis using Partek Genomics Suite and Galaxy data analysis software, an open source, web-based platform that provides tools necessary to create and execute RNAseq analysis. In brief, RNAseq data analysis pipeline was developed using Galaxy software workflow. Quality control (QC) check on the RNAseq raw reads was performed using the FastQC tool, followed by read trimming to remove base positions with a low median (or bottom quartile) score. Tophat2 Aligner tool mapped processed RNAseq reads to the hg19 human genome build. Picard’s CollectInsertSizeMetrics tool was applied on the initial tophat2 run to obtain estimated insert sizes, which was then used to calculate mean inner distance between mate pairs (mean = estimated_insert-size − 2 × read_length). Tophat2 was re-run using corrected mean value, and Cufflinks tool was used to assemble the reads into transcripts.

### 5.10. Bioinformatics Analysis

Gene expression data were filtered using the following criteria: genes with mean FPKM < 1 were removed. Global gene expression profile (GEP) data were analyzed further using a combination of R and Partek Flow to perform differential expression testing to identify GEP signatures of drug response. Mean fold change >j1j and *p* < 0.05 were considered thresholds for reporting significant differential gene expression. Differentially expressed gene analysis was performed between two groups of gene expression datasets (e.g., treated vs. untreated). Heatmaps were generated using unsupervised hierarchical clustering analysis based on the DEGs. Owing to the small sample size, Limma, an empirical Bayesian method, was used to detect DEGs, obtain *p*-values, and further provided a false discovery rate based on the *p*-value using the Benjamini–Hochberg procedure to detect the DEGs [50]. The advantage of using Limma compared with a traditional t-test is that it provides a moderated t-test statistic by shrinking the variance statistics and therefore improves the statistical power.

All samples were initially normalized to control day 0. Then, each MTD and EE timepoint was normalized to the corresponding control timepoint, e.g., day 7 MTD and EE samples were normalized to day 7 control. After normalization, the top 1000 genes with the lowest *p*-values were selected. Then, MTD and EE samples with a relative fold change difference less than 2 were removed. Finally, each gene required at least 2 timepoints with a fold change difference greater than 1.5 to remove one-off gene changes. Each gene was then manually investigated to determine its role and function using databases such as GeneCards as well as literature searches using PubMed [51]. Genes without a well-defined function or genes without a clear role were labeled as unknown and removed from the list (Figure 10). The complete criteria list without manual adjustments can be found in the Appendix A. Heatmaps were generated using heatmapper, a web-based tool [52].

### 5.11. Ingenuity Pathway Analysis (IPA)

IPA is a web-based software application that integrates and interprets the data derived from differential mRNA expression analysis. (IPA) software (QIAGEN, Venlo, Netherlands) was used to identify the most significantly affected (1) molecular pathways predicted to be activated or inhibited, (2) upstream regulator molecule such as miRNA, transcription factors, and (3) downstream effects and biologic processes that were increased or decreased, and (4) to predict causal networks, relationships, mechanisms, and functions relevant to changes observed in our dataset and (5) to perform predictive toxicology analysis using toxicogenomic approaches (IPA-Tox) [53].

### 5.12. scRNAseq

The presence of drug-resistant single-cell subpopulations (subclones) may have influenced differential responses to METRO therapy in PCa tumors. Therefore, we performed single-cell transcriptomics to identify resistant and sensitive subclones based on single-cell GEP signatures. Briefly, automated single-cell capture, and cDNA synthesis, were performed at ~5000 tumor cells/sample using 10X Genomics Chromium platform. Single-cell RNAseq-based gene expression analysis was performed on the Illumina HiSeq 2500 NGS platform (paired end. 2 × 125 bp, 100 cycles. v3 chemistry) at ~5 million reads per sample. scRNAseq data were analyzed using R, Seurat, and Partek Flow software packages. All statistical analyses were performed using the R statistical package, and GraphPad Prism with a two-sided *p*-value < 0.05 considered as statistically significant. Total sample numbers and replicates were determined by performing a power analysis with an effect size of 0.25 and a significance level of 0.05 with a power of 80%. IPA analysis was performed to identify regulators, relationships, mechanisms, functions, and pathways relevant to changes observed in our dataset.

### 5.13. Statistical Analysis

All statistical analyses were performed using R for statistical computing and graphics, v3.4.2, and GraphPad Prism v7.0. We used parametric methods to analyze differences between two groups of cells. If the assumption appeared violated, appropriate nonparametric procedures were used. All tests were two-sided, and differences with a *p* < 0.05 were considered statistically significant. The curve fitting and statistical analysis of the IC50 data were performed using Graphpad Prism (Dotmatics, Boston, MA, USA). The IC50 was determined at ½ of fitted maximal activity. Usually, an extra sum-of-squares F test was used to compare IC50 values between treated and control samples.

## Figures and Tables

**Figure 1 ijms-24-08490-f001:**
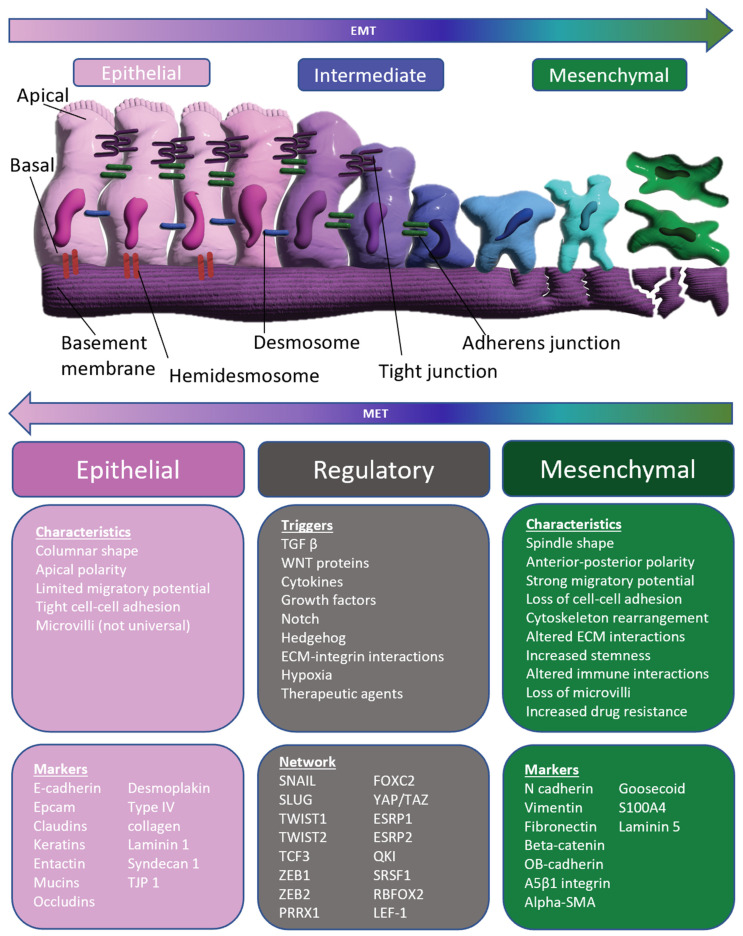
Overview of epithelial–mesenchymal transition (EMT) [13,16,17,18,19,20].

**Figure 2 ijms-24-08490-f002:**
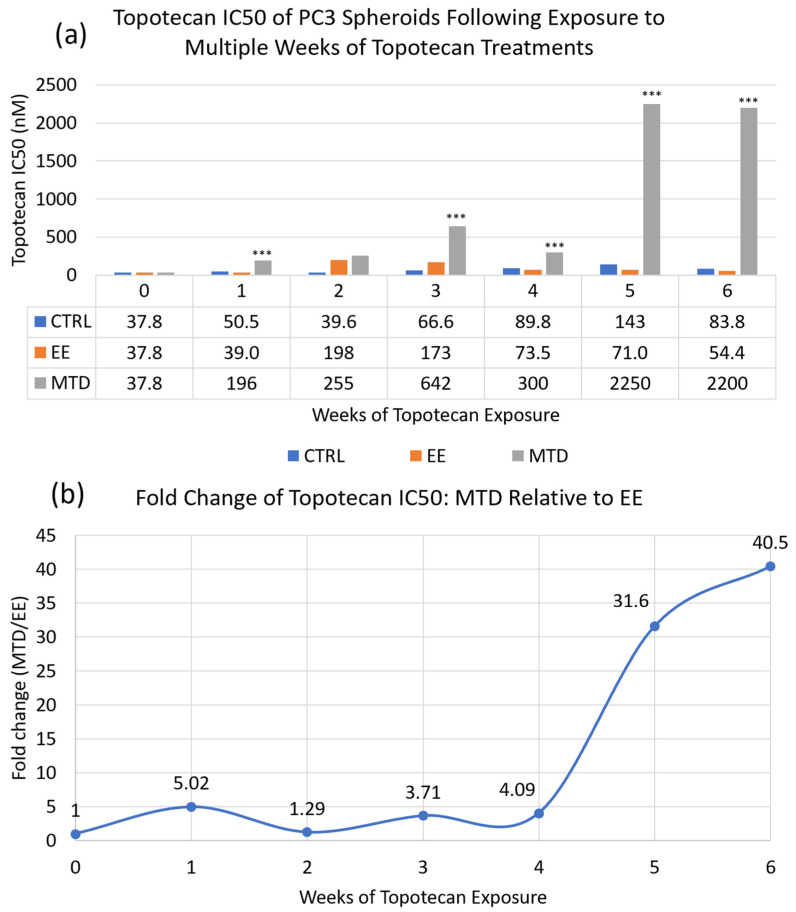
Long-term IC50 data for topotecan. (**a**) Column graph comparing the long-term IC50 of topotecan in a 3D spheroid model of PC3 cells following subsequent treatments. CTRL is control/untreated, EE is extended exposure topotecan/fractionated topotecan given at 1/7^th^ the MTD dose or 14.3 nM given daily, and MTD is maximum tolerable dosed topotecan/bolus topotecan given at the start of each week at 100 nM. (**b**) Relative fold change (MTD/EE) of the weekly IC50 data. After 6 weeks of treatment, control spheroids modestly reduced potency (37.8 nM to 83.8 nM), EE topotecan-exposed spheroids maintained similar potency (37.8 nM to 54.4 nM), and MTD topotecan-exposed spheroids significantly reduced potency (37.8 nM to 2200 nM). MTD topotecan-induced potency changes could be seen following the first week of exposure; however, these changes plateaued until another level of resistance occurred during weeks 5 and 6. *** Represents statistically significant differences (*p* < 0.05) between MTD and EE treatments.

**Figure 3 ijms-24-08490-f003:**
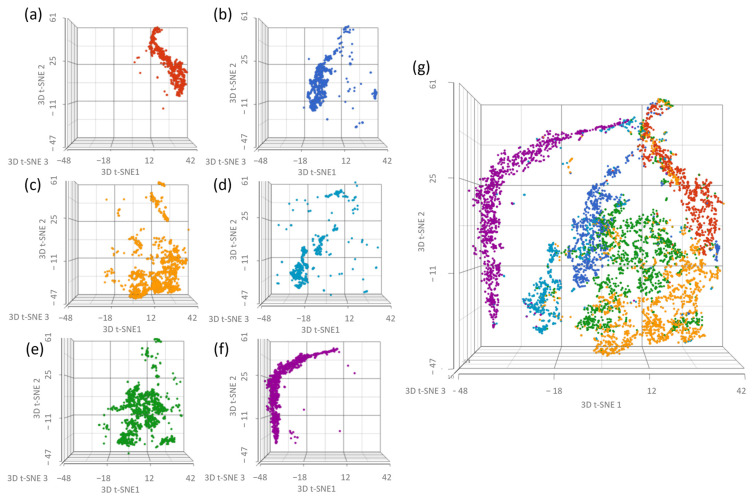
t-SNE analysis of scRNAseq data from pre-treatment 2D cells after 5 full weeks of treatment following a 2 week drug-free interval and from post-treatment 3D cells after the final day of the week 6 treatment. (**a**) CTRL week 5 no treatment 2D sample, (**b**) CTRL week 6 no treatment 3D sample, (**c**) MTD week 5 no treatment 2D sample, (**d**) MTD week 6 treatment 3D sample, (**e**) EE week 5 no treatment 2D sample, (**f**) EE week 6 treatment 3D sample, and (**g**) comprehensive analysis of all samples listed. Spheroids generated using the 5th week control cells showed increased heterogeneity relative to the 2D control cells. The 2D untreated EE and MTD samples were more heterogeneous than the 2D and 3D control samples and similarly heterogeneous to each other. However, the 2D EE sample was more similar to the CTRL sample than the 2D MTD sample. Most importantly, treatment with MTD topotecan increased the heterogeneity of the sample, and treatment with EE topotecan decreased the heterogeneity of the sample.

**Figure 4 ijms-24-08490-f004:**
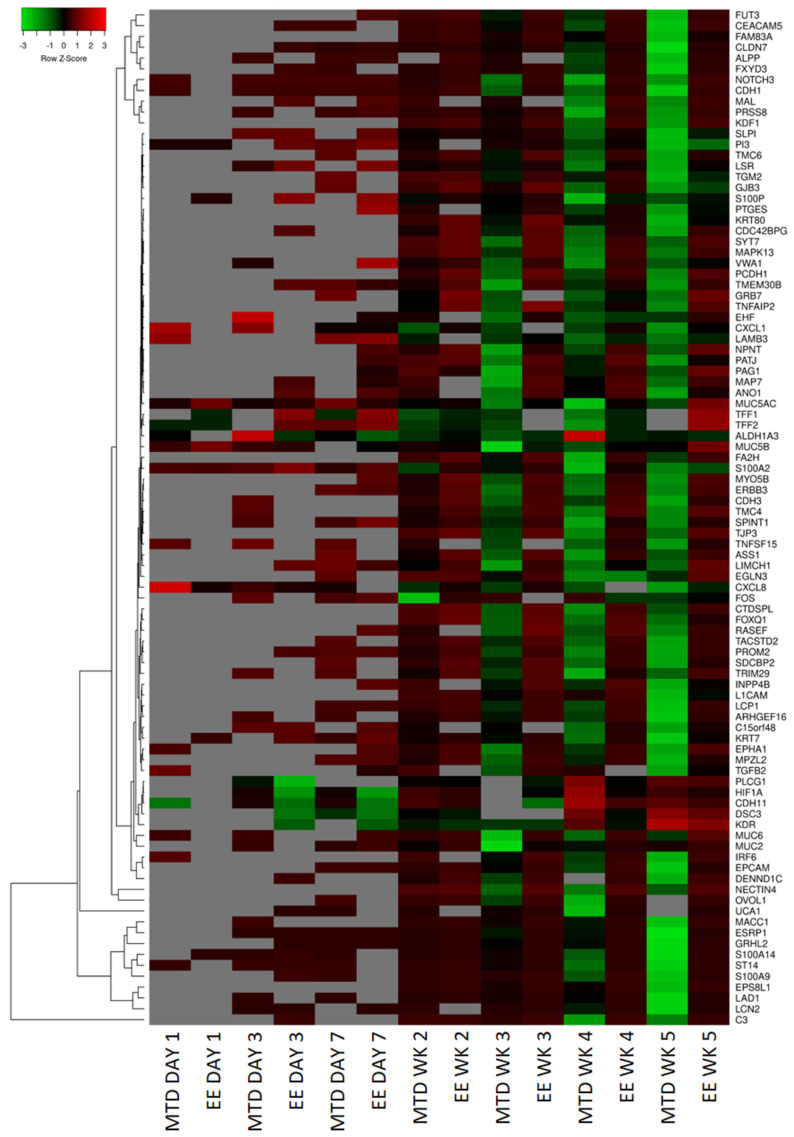
Heatmap of the top differentially expressed genes (RNAseq) over a five-week period from MTD- and EE-dosed spheroids. Grey boxes did not meet statistical significance. Green boxes are downregulated, red boxes are upregulated, and black boxes are expressed similarly to control. After 1 week of exposure, neither EE- nor MTD-treated cells differed significantly from control with the notable exception of *CXCL8*, which was highly expressed in MTD cells. However, it was not until the 3rd week of exposure that MTD-treated cells began to broadly change relative to control and EE-treated cells for this set of genes. These changes increased during weeks 4 and 5 with a greater number of genes showing changes and with increased fold changes relative to previous weeks. Comparatively, the genes of EE-treated cells were more stable over the study duration.

**Figure 5 ijms-24-08490-f005:**
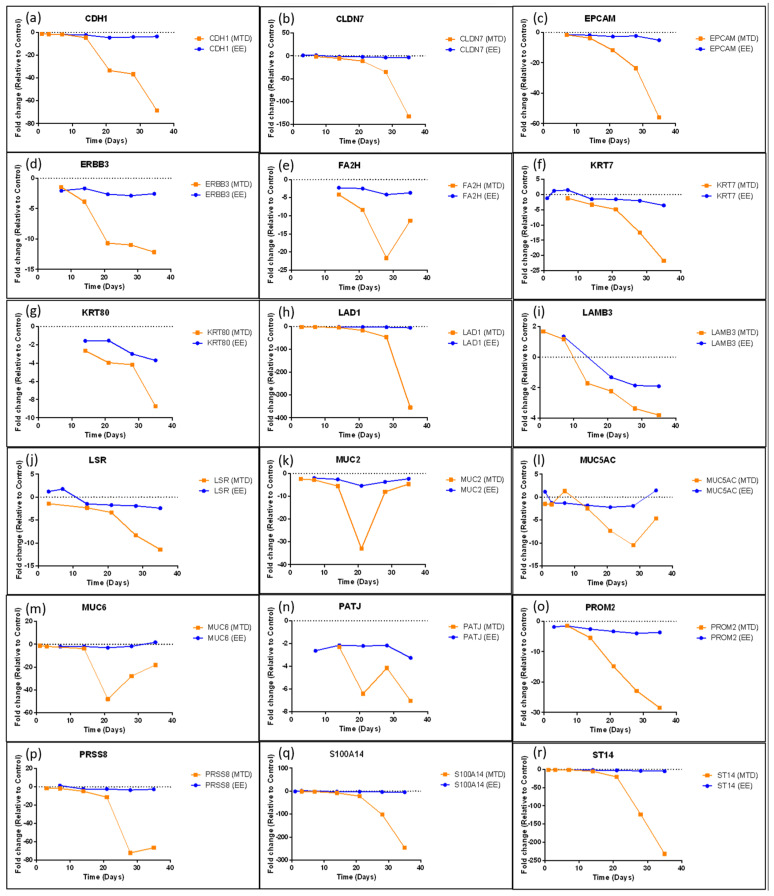
Long-term gene expression of select epithelial markers following five weeks of treatment with MTD or EE topotecan. (**a**) *CDH1*, (**b**) *CLDN7*, (**c**) *EPCAM*, (**d**) *ERBB3*, (**e**) *FA2H*, (**f**) *KRT7*, (**g**) *KRT80*, (**h**) *LAD1*, (**i**) *LAMB3*, (**j**) *LSR*, (**k**) *MUC2*, (**l**) *MUC5AC*, (**m**) *MUC6*, (**n**) *PATJ*, (**o**) *PROM2*, (**p**) *PRSS8*, (**q**) *S100A14*, (**r**) ST14. Spheroids exposed to MTD topotecan significantly reduced expression of many epithelial markers over the study period. These changes occurred roughly after the 3rd week of treatment for many genes and accelerated following weeks 4 and 5 of treatment. The genes of EE-treated spheroids showed relative stability with many genes maintaining similar expression patterns throughout the study. All data points must be statistically different from control for inclusion. Any missing data points indicate that the gene expression for that treatment and timepoint did not differ significantly from the control.

**Figure 6 ijms-24-08490-f006:**
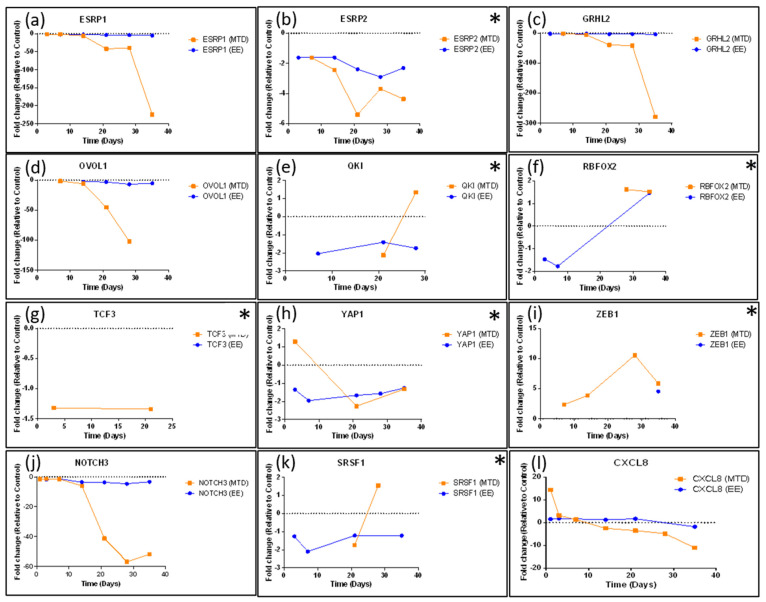
Long-term gene expression for select EMT regulatory genes following five weeks of treatment with MTD and EE topotecan. (**a**) *ESRP1*, (**b**) *ESRP2*, (**c**) *GRHL2*, (**d**) *OVOL1*, (**e**) *OKI*, (**f**) *RBFOX2*, (**g**) *TCF3*, (**h**) *YAP1*, (**i**) *ZEB1*, (**j**) *NOTCH3*, (**k**) *SRSF1*, (**l**) *CXCL8*. Many of known regulatory genes for EMT did not significantly differ consistently throughout the experiment, which can be seen in the graphs with missing datapoints. However, *ESRP1*, *ESRP2*, *GRHL2*, *ZEB1*, *NOTCH3*, and *CXCL8* were all significantly altered throughout the experiment. MTD topotecan induced significant downregulation in *ESRP1*, *GRHL2*, *OVOL1*, and *NOTCH3* and induced upregulation of *ZEB1*. *CXCL8* spiked initially after exposure to MTD topotecan but quickly normalized and was downregulated by week 5. EE topotecan-treated spheroids maintained similar expression throughout the study for most regulatory genes. * Represents genes that did not meet the initial screening criteria but were included to provide a more comprehensive view of the EMT transition. *SNAI1*, *SNAI2*, *TWIST1*, *TWIST2*, *ZEB2*, *PRRX1*, *FOXC2*, and *LEF1* were investigated for their importance in EMT but did not significantly differ from control at any timepoint. All data points must be statistically different from control for inclusion. Any missing data points indicate that the gene expression for that treatment and timepoint did not differ significantly from the control.

**Figure 7 ijms-24-08490-f007:**
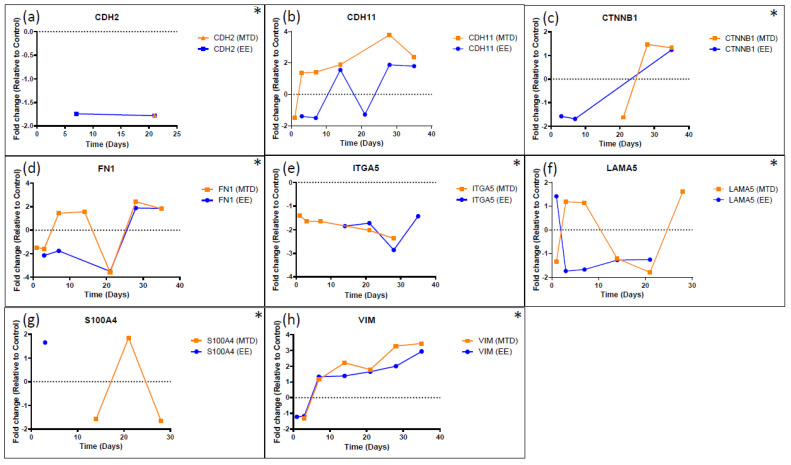
Long-term gene expression for select mesenchymal genes following five weeks of treatment with MTD and EE topotecan. (**a**) *CDH2*, (**b**) *CHD11*, (**c**) *CTNNB1*, (**d**) *FN1*, (**e**) *ITGA5*, (**f**) *LAMA5*, (**g**) *S100A4*, (**h**) *VIM*. Most of the genes shown above did not meet the initial screening criteria but were included to provide a comprehensive view of the EMT transition of EE- and MTD-treated spheroids. Only *CDH11* met the criteria, and although MTD-treated spheroids showed consistently higher expression compared to EE-treated spheroids, they did not drastically differ. * Represents genes that did not meet the initial screening criteria but were included to provide a more comprehensive view of the EMT transition. *ACTA2* and *GSC* were investigated for their role in EMT but did not significantly differ from control at any timepoint. All data points must be statistically different from control for inclusion. Any missing data points indicate that the gene expression for that treatment and timepoint did not differ significantly from the control.

**Figure 8 ijms-24-08490-f008:**
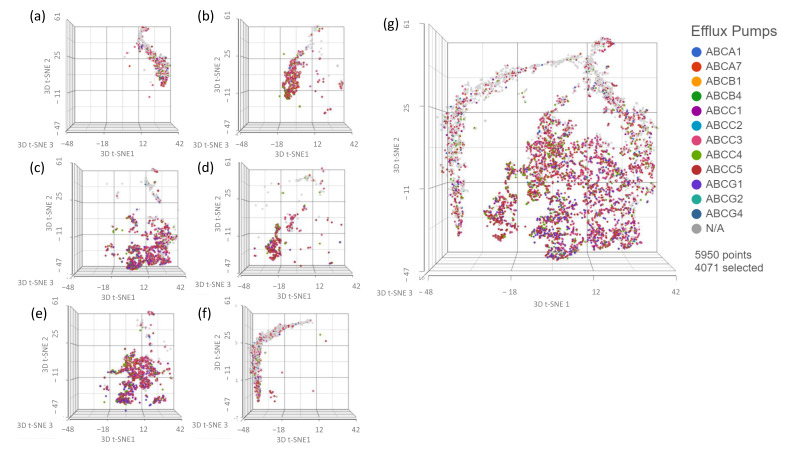
scRNAseq data demonstrating the effects of different treatments on efflux pump expression. (**a**) CTRL week 5 no treatment 2D sample, (**b**) CTRL week 6 no treatment 3D sample, (**c**) MTD week 5 no treatment 2D sample, (**d**) MTD week 6 treatment 3D samplI(**e**) EE week 5 no treatment 2D sample, (**f**) EE week 6 treatment 3D sample, and (**g**) comprehensive analysis of all samples listed. A larger proportion of cells from 3D CTRL spheroids expressed efflux pumps relative to the 2D CTRL spheroids. Cells from the EE- and MTD-treated 2D groups also showed an increased proportion of efflux pump expression as well as the expression of *ABCC5* and *ABCG1*, which was not highly expressed in the CTRL groups. Cells from the MTD-treated spheroids highly expressed efflux pumps; however, cells from the EE-treated spheroids did not. Instead, EE-treated spheroids demonstrated similar to reduced efflux pump expression to the untreated 2D CTRL group.

**Figure 9 ijms-24-08490-f009:**
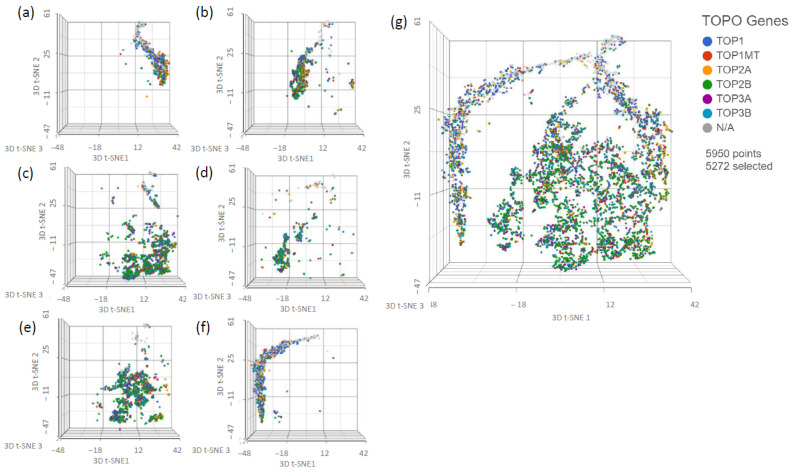
scRNAseq data demonstrating the effects of different treatments on topoisomerase expression. (**a**) CTRL week 5 no treatment 2D sample, (**b**) CTRL week 6 no treatment 3D sample, (**c**) MTD week 5 no treatment 2D sample, (**d**) MTD week 6 treatment 3D sample, (**e**) EE week 5 no treatment 2D sample, (**f**) EE week 6 treatment 3D sample, and (**g**) comprehensive analysis of all samples listed. Relative to the 2D CTRL sample, the 3D CTRL sample increased the proportion of *TOP2* to *TOP1*. Similarly, the EE and MTD 2D untreated samples increased the proportion of *TOP2* to *TOP1* but also increased the frequency of *TOP3* expression. The MTD-treated 3D sample also showed increased *TOP2* and *TOP3* expression relative to the *TOP1* expression; however, the EE-treated 3D sample did not show similar changes and instead appeared to express *TOP1* similarly to the 2D CTRL sample.

**Figure 10 ijms-24-08490-f010:**
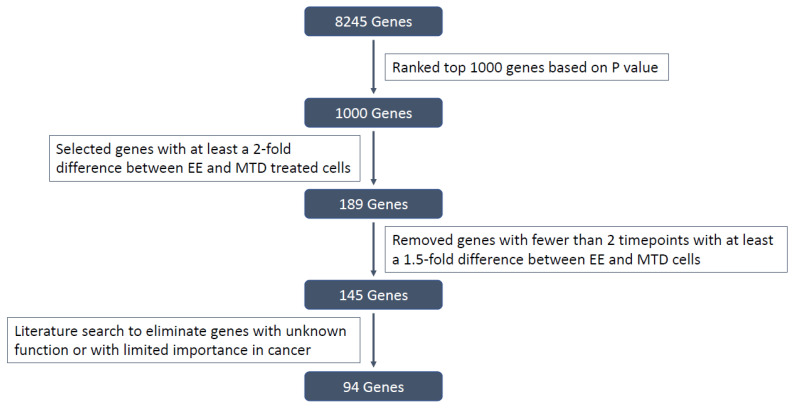
Illustration of the gene selection process and exclusion criteria.

## Data Availability

In accordance with AU Research Data Policy, the datasets generated during and/or analyzed during the current study are included in the article and the Appendix A, and are available from the corresponding author upon reasonable request.

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
