# Peer review of "Extended Exposure Topotecan Significantly Improves Long-Term Drug Sensitivity by Decreasing Malignant Cell Heterogeneity and by Preventing Epithelial–Mesenchymal Transition"

_ijms, 2023, doi:10.3390/ijms24108490_

Round 1

Reviewer 1 Report

The research article "Extended Exposure Topotecan Significantly Improves Long Term Drug Sensitivity by Decreasing Malignant Cell Heterogeneity and by Preventing Epithelial-Mesenchymal Transition" authored by Davis et al., is a comprehensive study. Here in this study authors investigated whether extended exposure (EE) of topotecan could improve long-term drug sensitivity by preventing drug resistance. The results presented in the article are clear and elaborate to establish the hypothesis proposed in the manuscript.

No Comments

Author Response

We thank you for taking the time to review our manuscript and for your positive comments.

Reviewer 2 Report

In their work, the authors presented a dosing strategy to overcome drug resistance in cancer cells and improve treatment efficacy. The work was performed on PC3 cancer spheroids treated for several days with the maximum tolerated dosage of topotecan on different protocols. RNAseq analysis was performed to distinguish between treatments. Differences were observed in efflux pumps and alternative topoisomerase genes depending on the protocol.

The data reported in the present study are interesting and promising for further investigation. The main comment relates to the presentation of the data. In the presentation of the t-SNE analysis, it is not possible to read the parameters indicated on the axes (Figure 3).

Figure 6 and Figure 7 show the expression of selected genes. Some of the plots do not include the blue data referring to the EE protocol (Figure 6g, 6i). Does this mean that the data are not shown?

** in the legends refer to some additional genes that are not indicated in the diagrams. However, the authors mention them in the text that appears in Figure 6, for example. Where are these data? The same applies to Figure 7.

Author Response

We thank you for your time and for your thoughtful feedback. We have incorporated the edits you suggested and clarified the text as described below.

Point 1: In the presentation of the t-SNE analysis, it is not possible to read the parameters indicated on the axes (Figure 3).

We have updated Figures 3, 8, and 9 to make the parameters and labels on the axes easier to read.

Point 2: Figure 6 and Figure 7 show the expression of selected genes. Some of the plots do not include the blue data referring to the EE protocol (Figure 6g, 6i). Does this mean that the data are not shown?

In Figures 5, 6, and 7, we required that each data point be statistically significant from the control. Any missing data points indicate that the gene expression of that gene for that time point and treatment was not statistically different from the control for the same time point. We have updated those figures to make this more explicit. The following text has been added to each of the figure legends.

“All data points must be statistically different from control for inclusion. Any missing data points indicate that the gene expression for that treatment and timepoint did not differ significantly from the control.”

Point 3: ** in the legends refer to some additional genes that are not indicated in the diagrams. However, the authors mention them in the text that appears in Figure 6, for example. Where are these data? The same applies to Figure 7.

Regarding the other genes listed in Figures 6 and 7, we wanted to ensure a complete overview of EMT and included other genes that did not explicitly meet our screening criteria to be thorough. However, for the genes listed, gene expression did not significantly differ from control for any time point or treatment. Because of this, these data points would not have been plotted and we didn’t want to include empty graphs.

Reviewer 3 Report

In the manuscript „Extended Exposure Topotecan Significantly Improves Long Term Drug Sensitivity by Decreasing Malignant Cell Heterogeneity and by Preventing Epithelial-Mesenchymal Transition“ Davis et al. show how dosing of chemotherapeutic drugs, such as topotecan, can influence the characteristics of the surviving cancer cell subpopulation. The authors used 2D and 3D model of PC3 prostate cancer cell line to determine the potency of topotecan long-term treatment with either MTD (maximum tolerable dosing) or EE (extended exposure) dosing strategy. The authors reported increased IC50 value of MTD-topotecan treated cells compared to EE-topotecan cells that retained topotecan sensitivity of PC3 cells. Furthermore, in-depth transcriptomic analysis revealed that MTD-topotecan treated cells undergo changes associated with chemotherapy resistance, e.g. experience shift similar to EMT, upregulate expression of diverse ABC efflux pumps and topoisomerases. On the other hand, EE-topotecan treated cells display reduced malignant properties and maintain treatment response. Based on their findings authors conclude that drug dosing can impact surviving cancer cell subpopulations, affect drug efficacy and potentially treatment outcome.

I only have few comments where one concerns the cancer type and treatment regiment. Could authors please explain what was a rationale of using topotecan treatment for prostate cancer since this type of chemotherapy is not used in the clinic for the treatment of PC? It would be interesting to check whether docetaxel dosing (MTD vs EE) would also influence the characteristics of PC cells.

The other minor remark concerns reference citation/numbering in the row 477, where no. 51 should be replaced with 53, since no. 53 in the reference list refers to Metzger et al.

Nevertheless, the topic of the manuscript is very interesting and, in my opinion, is of immense scientific interest. The manuscript is written in a clear way, shows intriguing results and will contribute greatly to the field.

Author Response

Thank you for your time and for your thoughtful comments and suggestions.

Point 1: Could authors please explain what was a rationale of using topotecan treatment for prostate cancer since this type of chemotherapy is not used in the clinic for the treatment of PC? It would be interesting to check whether docetaxel dosing (MTD vs EE) would also influence the characteristics of PC cells.

Initially, we investigated whether extended exposure (EE) dosing of chemotherapeutics could improve efficacy in prostate cancer, we screened several different agents such as docetaxel, paclitaxel, vincristine, doxorubicin, cyclophosphamide, and topotecan. Importantly, EE docetaxel did not appear to show any improvement in activity relative to the MTD dose and in many cases appeared to be slightly less effective. Of the agents tested, extended exposure topotecan appeared to show the greatest activity in prostate cancer, so we selected this agent for further investigation. MTD topotecan was initially investigated in prostate cancer as a phase 2 clinical trial but did not significantly improve therapy and was abandoned. We believe that our data might explain why MTD topotecan ultimately failed clinically and why EE dosing schedules of topotecan may be successful. We ultimately see this therapy as a potential adjunct to docetaxel and have investigated combination therapy (in another paper) in prostate cancer using docetaxel and topotecan and have found a synergistic response with EE topotecan, but not with MTD topotecan. Another manuscript that focuses on different drug-drug combinations and dosing schedules is in preparation. We choose to separate that data from this paper to focus on the impact of EE on EMT and drug resistance.

Point 2: The other minor remark concerns reference citation/numbering in the row 477, where no. 51 should be replaced with 53, since no. 53 in the reference list refers to Metzger et al.

We have updated the citation in row 477. Thank you for identifying that error.